# Corneal Biomechanical Measures for Glaucoma: A Clinical Approach

**DOI:** 10.3390/bioengineering10101108

**Published:** 2023-09-22

**Authors:** Abdelrahman M. Elhusseiny, Giuliano Scarcelli, Osamah J. Saeedi

**Affiliations:** 1Department of Ophthalmology, Harvey and Bernice Jones Eye Institute, University of Arkansas for Medical Sciences, Little Rock, AR 72205, USA; amelhusseiny@uams.edu; 2Department of Ophthalmology, Boston Children’s Hospital, Harvard Medical School, Boston, MA 02114, USA; 3Fischell Department of Bioengineering, University of Maryland, College Park, MD 20742, USA; scarc@umd.edu; 4Department of Ophthalmology and Visual Sciences, University of Maryland School of Medicine, Baltimore, MD 21201, USA

**Keywords:** glaucoma, corneal biomechanics, ocular biomechanics, corneal hysteresis, ocular response analyzer, Corvis ST, Brillouin

## Abstract

Over the last two decades, there has been growing interest in assessing corneal biomechanics in different diseases, such as keratoconus, glaucoma, and corneal disorders. Given the interaction and structural continuity between the cornea and sclera, evaluating corneal biomechanics may give us further insights into the pathogenesis, diagnosis, progression, and management of glaucoma. Therefore, some authorities have recommended baseline evaluations of corneal biomechanics in all glaucoma and glaucoma suspects patients. Currently, two devices (Ocular Response Analyzer and Corneal Visualization Schiempflug Technology) are commercially available for evaluating corneal biomechanics; however, each device reports different parameters, and there is a weak to moderate agreement between the reported parameters. Studies are further limited by the inclusion of glaucoma subjects taking topical prostaglandin analogues, which may alter corneal biomechanics and contribute to contradicting results, lack of proper stratification of patients, and misinterpretation of the results based on factors that are confounded by intraocular pressure changes. This review aims to summarize the recent evidence on corneal biomechanics in glaucoma patients and insights for future studies to address the current limitations of the literature studying corneal biomechanics.

## 1. Introduction

Glaucoma is a heterogeneous group of disorders characterized by progressive optic neuropathy with subsequent visual field defects if left untreated. It is a leading cause of irreversible vision loss worldwide, with an estimated global prevalence of 2.4–3.5% in people above the age of 40 and a higher prevalence in African origin populations [1,2]. Although intraocular pressure (IOP) reduction is the only proven modifiable risk factor in reducing the glaucoma progression [3], 30–57.1% of primary open-angle glaucoma (POAG) patients have normal IOP (normal tension glaucoma, NTG) [4,5,6,7]. Furthermore, ocular hypertension (OHT) patients may never develop glaucomatous damage [8]. Other risk factors such as thinner corneas, high myopia, genetic susceptibility, vascular factors, and family history have been postulated to contribute to glaucoma development and progression [8,9,10].

From a biomechanical perspective, the stress (applied force) of IOP does not cause glaucomatous damage itself, but rather the resulting strain (deformation) to the ocular tissues, specifically to the peripapillary sclera and the optic nerve head, where retinal ganglion cell (RGC) axons are most vulnerable; thus, determination of the modulus of elasticity (i.e., resistance to tissue deformation under applied stress) is critical to understanding the impact of IOP on ocular tissues and ultimately patient-specific susceptibility to the development and progression of the disease [11,12,13]. While the direct determination of the peripapillary scleral modulus is impractical clinically, given the overlying conjunctiva and posterior location, the corneal modulus is routinely measured in clinical settings to aid in the measurement of IOP and, more recently, to assist in glaucoma diagnosis and management [11,12,13]. Furthermore, the corneal biomechanical properties indicate the corneal capacity to dissipate energy from routine changes in IOP that may result from eye movement, blinking, or head movement. Hence, altered or impaired corneal biomechanical properties could increase the exposure of the optic nerve to IOP fluctuation and ultimately result in greater susceptibility to glaucomatous damage [14,15].

Several devices have been developed for in vivo evaluation of corneal biomechanics. These devices include the Ocular Response Analyzer (ORA; Reichert, NY, USA), the Corneal Visualization Scheimpflug Technology (Corvis ST, Oculus, Wetzlar, Germany), and, more recently, Brillouin microscopy (BM). Each device provides distinct sets of parameters for assessing corneal biomechanics. The ORA primarily measures corneal hysteresis (CH) and corneal resistance factor (CRF). These values are calculated based on applanation pressure data obtained when an air jet is applied to the cornea [16]. The CH has shown promise as a predictive factor for glaucoma development in individuals at risk of the disease: “glaucoma suspects”. Moreover, CH has been associated with the rates of glaucoma progression and visual field loss in diagnosed glaucomatous patients [16]. 

In contrast, Corvis ST offers a more detailed evaluation of corneal biomechanical properties. It achieves this by directly visualizing corneal deformation and geometrical changes caused by the air jet. The latest Corvis ST software (1.3r1538) provides information on approximately 37 parameters [17]. An emerging method, BM, offers a non-contact three-dimensional evaluation of corneal biomechanics. It relies on the light scattering and is independent of the IOP. However, its application in evaluating corneal biomechanics in glaucoma patients remains unexplored. 

The purpose of this review is to provide an overview of the current modalities assessing corneal biomechanics, the evidence of their relevance to glaucoma, and challenges in assessing corneal biomechanics in glaucoma patients that may help direct future research in this field.

## 2. Foundational Concepts in Corneal Biomechanics

A foundation of biomechanical principles and definitions is essential for a better understanding of the parameters reported by each device. Stiffness is the measure of the resistance of a specific material to being deformed when a certain force is applied to it [18]. As more than 90% of the cornea is composed of stacked collagen fibrils lamellae within the corneal stroma, the corneal stiffness is affected by collagen fibers’ thickness and density. Factors such as age, diabetes, cross-linking, and glaucoma affect collagen density and alter corneal stiffness [19]. Furthermore, corneal biomechanics is most frequently described in the context of the stress–strain relationship. Stress is the force applied to a specific area (stress = force/cross-sectional area) and describes the inner resistance of the material when deformed. Strain is the deformation resulting from the applied force (stress) (strain = elongation (difference in length upon deformation/original length). In the eye, the IOP exerts pressure on the inner structures, including the cornea and lamina cribrosa, creating stresses throughout the corneal thickness and in multiple directions [18].

In the case of linearly elastic materials, the relationship between stress and strain is linearly proportional. The stress–strain slope defines the elastic modulus (Young’s modulus), so the higher the slope, the more force is needed to deform a stiffer material [20]. However, the elastic modulus of the cornea is non-linear, and the J-shaped stress–strain curve is different from that of linear elastic material [21]. 

Furthermore, IOP is a major confounding factor where the higher the IOP, the stiffer the cornea. For example, an originally softer cornea will exhibit a stiffer behavior in the presence of a high IOP than an originally stiffer cornea at a lower IOP. This highlights the importance of accounting for the role of IOP in the determination of corneal biomechanics [21,22]. 

The cornea also is characterized by being anisotropic and viscoelastic. Anisotropy is described when a substance has mechanical properties depending on the direction of the applied force. In other words, the biomechanical measures tested differ along different corneal meridians [18]. Material elasticity is described as the ability of the substance to return to its original form in a way similar to the deformation when the force was applied. However, viscosity means that part of the applied force is lost to internal friction as heat [18]. The cornea is a viscoelastic material, and part of the applied stress is lost. Therefore, the deformation during the loading phase differs from that during the unloading phase, and the difference is defined as mechanical CH. As a result of its viscoelastic properties, the corneal biomechanical response differs based on the rate of the applied stress, so the faster the rate of the applied force, the stiffer the corneal biomechanical response. In clinical settings, if the IOP is measured by air puff, the result will depend on the rate at which the air jet is applied [18].

## 3. In Vivo Clinical Assessment of Corneal Biomechanics

Evaluation of corneal biomechanics is challenging given the non-linearity of the corneal elastic modulus, anisotropy, and viscoelastic characteristics of the cornea. Several devices have been developed to evaluate corneal biomechanics; however, each has advantages and limitations.

### 3.1. Ocular Response Analyzer

In 2005, the ORA was introduced in clinical practice as the first device for evaluating corneal biomechanical behavior in vivo [23]. It is a non-contact tonometer that uses an air jet applied to the cornea’s central 3–6 mm, causing corneal deformational changes. Those bi-directional changes are monitored by an advanced infrared electro-optical system that detects the corneal surface’s infrared reflection during its deformation. Once the cornea is applanated (first applanation event) in response to the air jet, the piston releasing the air shuts down, allowing the cornea to return to its original form. The pressure at which the cornea applanates is defined as P1. However, due to the piston inertia, the air pressure continues to increase to its highest level (Pmax), causing further indentation of the cornea, which becomes slightly concave. As the piston produces the air shut-off, the air pressure subsequently decreases. Hence, the cornea returns to its original shape, gradually passing through a second applanation event before returning to its initial convex configuration. The air pressure at the second applanation event is defined as P2. Due to the corneal viscoelastic properties, the P2 (unloading pressure) value is always smaller than the P1 (loading pressure) value, and the difference between both is known as CH (measured in mmHg) (Figure 1). 

The Goldmann-correlated IOP (IOPg) measured by the ORA was the average of P1 and P2 values. In contrast, the corneal-compensated IOP (IOPcc) was developed through empirical investigation to compensate for corneal factors in measuring the IOP, presumably producing more accurate IOP measurements, especially after refractive surgery [24]. Another parameter that the ORA reports is the CRF. The CRF is calculated based on this equation: CRF = a (P1 − bP2) + d, where a, b, and d are constants [20,25,26]. The CRF was developed to maximize correlation with the central corneal thickness (CCT). It should be noted that all four parameters reported by the ORA are calculated based on P1 and P2.

Previous studies have shown that CH is affected by age, CCT, diabetes status, keratoconus, and glaucoma [27,28,29]. An ex vivo study on rabbit eyes by Bao and colleagues found IOP to be highly correlated with CRF and weakly correlated with CH [30]. Another study by Touboul and colleagues concluded that CH was moderately dependent on IOP and CCT [27]. Although several studies have evaluated the corneal biomechanics using the ORA in different diseases, a lot of those studies did not account for the confounding nature of IOP and CCT on the reported ORA parameters, which makes it hard to interpret the results and draw a solid conclusion [20].

### 3.2. Corneal Visualization Scheimpflug Technology (Corvis ST)

In 2010, Oculus introduced Corvis ST as a non-contact method for in vivo assessment of corneal biomechanics. It is based on combining bidirectional dynamic applanation as in ORA and recording the deformational corneal changes through an ultra-high-speed Scheimpflug camera. A single concentric air jet is applied to the cornea, which is subsequently deformed, starting with an inward deformation phase, then the applanation phase, and then the highest concavity phase before returning to its original shape and passing through a second applanation phase. The ultra-high speed (4300 frames/second) Scheimpflug camera takes 140 images of the horizontal corneal meridian during the 32 milliseconds duration of the air jet. The images are further analyzed to report the dynamic corneal response parameters (DCR). Although both ORA and Corvis ST use air puffs, there is a difference between them. In the ORA, the air jet is variable depending on the P1, while the air pressure is constant in the Corvis ST [18,25,31,32].

The Corvis ST reports several parameters (Figure 2); however, it may also be confounded by the IOP [20,26,31]. 

The applanation times (AT), lengths (AL), and velocities (AV) are recorded during the inward (first applanation, designated #1, e.g., A1T, A1L, A1V) and outward (second applanation, designated #2, e.g., A2T, A2L, A2V) phases of corneal applanation. The maximum corneal deformation during the air puff is known as maximum deformation (in millimeters). The curvature radius highest concavity (radius HC) is the corneal radius of curvature in millimeters at the highest corneal concavity, whereas the maximum inverse radius is 1/radius HC. It should be noted that the higher the radius HC, the more resistance to deformation, i.e., stiffer cornea, whereas the higher the inverse radius, the less resistance to deformation, i.e., softer cornea. The integrated inverse radius (IIR) is the integrated sum of the inverse concave radius between the first and second applanation [26,32]. During the air puff, a whole-eye movement (WEM) occurs. The parameters that compensate for WEM are known as “deflection” parameters, whereas those that do not compensate for WEM are “deformation” parameters. For example, the maximum deformation amplitude (DA Max) is the displacement of the corneal apex in the anterior–posterior plane. 

In contrast, the maximum deflection amplitude (DeflAmpMax) is the DA Max minus WEM [26,32]. The DA ratios 1 or 2 mm are the DA Max divided by the DA at 1 or 2 mm away from the apex. The higher the DA ratio, the softer the cornea because the deformation occurs at the center but is limited at the periphery in a softer cornea. The peak distance (PD) is the distance between the two corneal peaks at the time of the highest corneal concavity. The Ambrosio Relational Thickness to the horizontal profile (ARTh) is the quotient corneal thickness at the thinnest point of the horizontal meridian, and the thickness changes [17,26,32,34]. The corneal biomechanical index (CBI) was developed using statistical methods to enhance keratoconus detection and screening [35]. Biomechanically corrected IOP (bIOP) compensates for the effects of CCT on IOP measurements. However, a recent study by Matsuura and colleagues demonstrated that bIOP was significantly associated with CH (*p* < 0.001). On the other hand, IOPcc measured by the ORA was not significantly associated with CH but was significantly associated with CCT [36].

Recently, more parameters have been developed for a more accurate evaluation of corneal stiffness, including the Stiffness Parameter at the first Applanation (SP-A1), Stiffness Parameter at Highest Concavity (SP-HC), and Stress–Strain Index (SSI). Those parameters have been proposed to be heavily affected by corneal stiffness rather than the IOP [31]. The SSI was developed in 2019 by Eliasy and colleagues for mapping the overall corneal stiffness [37]. Zhang and colleagues have reported that the SSI map values demonstrated small fluctuations with IOP and CCT [38]. Pillunat and colleagues developed a novel parameter called biomechanical glaucoma factor (BGF) based on several Corvis ST parameters, including DA ratio progression, HCT, Pachymetry slope, bIOP, and Pachymetry [39]. Although a study by Fujishiro and colleagues demonstrated a significant correlation between Corvis ST measurements (DA ratio, SP-A1, and inverse radius) and ORA-measured CH, the correlation was weak to moderate. The authors have suggested that the optimal model for calculating CH using Corvis ST parameters is: CH = −76.3 + 4.6xA1T + 1.9xA2T + 3.1xDA + 0.016xCCT [17]. 

### 3.3. Brillouin Microscopy

The BM is a non-contact device that uses a different approach for evaluating corneal biomechanics based on light scattering (Brillouin scattering) arising from the interaction between light photons and acoustic phonons (thermodynamic fluctuations). Upon Brillouin interaction, the scattered light acquires a frequency shift related to the longitudinal elastic modulus of the sample without needing any mechanical perturbation [40]. Unlike previous methods, BM provides a non-contact, non-perturbative three-dimensional (3D) mapping of the corneal elastic modulus. Furthermore, its novel elasticity metrics enable distinguishing ectatic from normal corneas in vivo with previously unattainable mechanical sensitivity [41,42]. The measurement, as currently performed, is also independent of IOP [43] and thus may help solve the problem of the confounding effect of IOP on corneal biomechanical measures. This is particularly an issue in glaucoma, where glaucoma patients may have high IOP or lower treated IOP.

In vivo corneal biomechanics have never been evaluated in glaucoma patients using BM. However, several studies have used BM to evaluate corneal biomechanics in keratoconus, collagen cross-linking efficacy, and in vivo crystalline lens evaluation [44,45,46,47]. In an ex vivo study, Scarcelli and colleagues reported that the keratoconic corneas had a significantly lower Brillouin frequency shift in the cone area compared to normal corneas (*p* < 0.001) [44]. However, there were no significant differences in the mean Brillouin frequency in an area outside the cone compared to corresponding areas in normal healthy corneas. In the most recent in vivo studies by Zhang and colleagues, motion-tracking was introduced to enhance Brillouin measurement sensitivity [42]; they retrospectively compared the corneal biomechanics of early keratoconus patients to healthy controls. They reported a statistically significant reduction in the Brillouin frequency shift of keratoconic corneas compared to normal corneas (*p* < 0.001), demonstrating the great potential of mechanical metrics to identify the earliest stage of ectasia progression [41]. Further studies are needed to evaluate the utility of the BM in other disease conditions, including different types of glaucoma [46].

Other methods using ultrasound or optical coherence tomography (OCT) are being developed to evaluate corneal biomechanics, such as ultrasound elastography, OCT elastography, and electronic speckle pattern interferometry [25,48].

## 4. Clinical Studies Measuring Corneal Hysteresis and Corneal Resistance Factor in Glaucoma Patients

Several studies have evaluated the ORA parameters (CH and CRF) in different types of glaucoma and OHT (Table 1) [49,50,51,52,53,54,55,56,57,58,59,60,61,62,63,64,65,66,67,68,69,70,71,72,73,74].

Most studies have reported that CH is lower in glaucoma/OHT patients compared to healthy controls. This may reflect corneal biomechanical differences in glaucoma but may be confounded by IOP. For example, in cases of increased IOP, the tension on the cornea increases, and its ability to dissipate energy decreases, resulting in smaller differences between the P1 and P2 and, accordingly, a lower CH [20]. 

Several factors must be considered in interpreting the results of these studies. A common misconception is to interpret CH/CRF values as parameters for corneal stiffness, although both are parameters for elasticity and viscosity rather than purely elasticity parameters. In other words, low CH by itself does not mean a soft or stiff cornea [15,20]. Second, some studies have evaluated CH/CRF in POAG without stratifying them into high-tension glaucoma (HTG) and NTG, which have different biomechanical profiles [50,51,54]. Age and diabetes status have further been reported to affect CH; therefore, any interpretation of the results should consider adjusting for those factors [75,76].

The CH and CRF have been reported to differ among different types and stages of glaucoma. In a study of 894 subjects, Rojananuangnit retrospectively compared CH in glaucoma patients (POAG-HTG, POAG-NTG, primary angle-closure glaucoma (PACG), and OHT) to normal controls. He reported that the mean CH was significantly lower in POAG-HTG compared to POAG-NTG and OHT. However, the difference was not statistically significant between POAG-HTG and PACG. In POAG-HTG and PACG, mean CH was significantly different between different stages of glaucoma, being lower in more severe stages of the disease. For example, the mean CH in the POAG-HTG severe stage was 7.92 mmHg compared to 9.22 mmHg in the early stage and 8.74 in the moderate stage (*p* < 0.001). In PACG, the mean CH was statistically significantly lower in the severe stage (8.45) compared to the early stage (9.85) (*p* = 0.004) but not when compared to the moderate stage (9.04, *p* = 0.2) [71]. In contrast, in a study of 49 patients, Yang and colleagues compared the CH in POAG-HTG versus POAG-NTG and reported no significant difference (10.11 mmHg versus 10.17 mmHg, respectively) (*p* = 0.81) [77]. A cross-sectional study of 162 subjects by Beyazyildiz and colleagues found that the mean CH was significantly lower in pseudoexfoliative glaucoma (PXG) (7.6 mmHg) compared to POAG patients (9.1 mmHg) and normal controls (9.6 mmHg) (*p* < 0.001). The CRF was also significantly lower in PXG patients (9.0 mmHg) compared to POAG patients (10.1 mmHg) and healthy controls (9.8 mmHg) [60].

Other studies have investigated the relationship between CH and glaucomatous structural changes. In a multicenter prospective study (EPIC-Norfolk Eye Study), Khawaja and colleagues evaluated the association between CH and Heidelberg retina tomograph (HRT) and Glaucoma Detection with Variable Corneal Compensation scanning laser polarimeter (GDxVCC). They reported that the CH was positively correlated with HRT rim area and GDxVCC-derived retinal nerve fiber layer (RNFL) thickness and modulation and negatively correlated with the HRT-derived linear cup-to-disc ratio [78]. Another prospective study by Wells and colleagues found that the CH was significantly correlated with the mean cup depth in glaucoma patients [79]. 

Further work has studied the potential association between CH and structural glaucoma progression. Wong and colleagues demonstrated that lower CH is significantly associated with anterior lamina cribrosa displacement, suggesting lower CH could be a risk factor for glaucoma progression [80]. Jammal and Medeiros measured the neuroretinal rim by the OCT of the Bruch’s membrane opening minimum rim width (MRW) and correlated it with baseline CH in 118 glaucomatous eyes. They reported that lower baseline CH was associated with faster loss of neuroretinal rim and that for each one mmHg lower baseline CH, the MRW loss was faster by −0.38 µ/year [81]. Radcliffe and colleagues reported that eyes with optic disc hemorrhage—another potentially important sign of glaucoma progression—have significantly lower CH (8.7 mmHg) compared to those without disc hemorrhage (9.2 mmHg) (*p* = 0.002) [82].

Finally, lower CH has been associated with a higher chance of visual field progression. Medeiros and colleagues conducted a prospective longitudinal study, including 114 glaucomatous eyes, with a mean follow-up of 4 years to demonstrate the role of baseline CH on visual field progression. They reported that the visual field index declined at a 0.25% faster rate annually per one mmHg lower CH [83]. Another prospective study by Kamalipour and colleagues included 248 glaucomatous and glaucoma suspect eyes with a mean follow-up of 4.8 years. They reported that for each one mmHg lower baseline CH, there was a faster decline in the 10-2 visual field mean deviation (MD) (0.07 dB/year) and 1.35 increased odds of visual field progression. However, there was no statistically significant correlation between the CH and 24-2 visual field MD [84]. Chan and colleagues reported that for each one mmHg decline in the CRF over time, the visual field MD declined by a 0.14 dB faster rate annually (*p* = 0.007) [85]. A recent study reported that lower baseline CH was associated with more rapid rates of optic nerve microvasculature loss in POAG patients [86].

Lower CH has been proposed as a risk factor for the development of glaucoma in glaucoma suspects. In a prospective study by Susanna and colleagues following up 287 eyes identified as glaucoma suspects, 44 eyes developed visual field defects during the follow-up. They demonstrated that baseline CH was significantly lower in patients who developed glaucoma (9.5 mmHg) compared to those who did not develop glaucoma (10.2 mmHg) (*p* = 0.01). They further demonstrated a 21% increased glaucoma risk for each one mmHg lower baseline CH [87].

## 5. Clinical Studies Using Corneal Visualization Scheimpflug Technology (Corvis ST) in Evaluating Corneal Biomechanics in Glaucoma Patients

While several studies have reported promising results for corneal biomechanical biomarkers using Corvis ST in glaucoma patients, the results have shown differing and sometimes conflicting results. Several studies have reported that glaucoma patients have less deformable corneas than normal controls [88,89,90], whereas other studies reported the reverse—more deformable corneas in glaucoma patients compared to controls [91,92]. This discrepancy may be related to certain limitations in the development of the technology as well as in the study design. For example, some studies drew conclusions based on corneal stiffness parameters that may be more dependent and confounded by IOP. Other limitations in the current literature include the inclusion of patients on chronic prostaglandin analogue (PGA) therapy, which may alter corneal biomechanics. Topical PGA alters the expression of matrix and tissue metalloproteinases, causing structural changes that may affect corneal stiffness and biomechanical properties. Finally, glaucoma at high pressures (HTG) and glaucoma at normal pressures (NTG) may have different biomechanical properties, and some prior work has not stratified POAG patients in one classification or another [14,31]. A recent meta-analysis by Catania and colleagues included six prospective studies comparing the Corvis ST parameters between POAG-HTG versus normal controls. They concluded that POAG-HTG patients had stiffer corneas than normal controls based on significantly lower DA, PD, HCT, A1V, and A2T and significantly higher radius HC compared to healthy controls [14]. 

It should be noted that mostof these factors were correlated strongly with IOP [31]. On the other hand, factors such as SP-A1, SP-HC, or SSI, which were less affected by IOP, were not included in their analysis [14]. Table 2 summarizes major studies evaluating corneal biomechanics in glaucoma/OHT patients using Corvis ST [33,39,70,88,89,90,91,92,93,94,95,96,97,98,99,100,101,102,103,104,105,106,107].

Few studies compared the corneal biomechanics of OHT patients versus POAG patients using Corvis ST. Silva and colleagues demonstrated that OHT eyes had less deformable “stiffer” corneas based on significantly higher SP-A1 compared to POAG patients (*p* = 0.04), although subjects were not stratified into HTG and NTG [103]. On the other hand, a study by Vinciguerra and colleagues reported that NTG eyes had more deformable “softer” corneas compared to those with OHT and HTG based on significantly lower SP-A1, SP-HC, and higher DA ratio and inverse concave radius [97].

BGF is a summary metric developed by Pillunat and colleagues, composed of several Corvis ST parameters. They proposed that a cut-off value of 0.5 may help differentiate NTG eyes from normal healthy control, with an area under the curve (AUC) of 0.8 and a sensitivity of 76% [39]. However, a larger study by Aoki and colleagues reported that BGF was not a helpful parameter for differentiating POAG and normal controls with an AUC of 0.61 [70]. In their cohort, they reported a higher AUC (0.7) for ORA-measured CH [70]. However, they did not differentiate between HTG and NTG and included patients on topical glaucoma medications. BGF has also been combined with anterior chamber parameters in evaluating PACG patients in a retrospective study that showed that median BGF was significantly lower in the PACG patients (6.2) compared to controls (6.6) (*p* < 0.001). The anterior chamber volume and BGF combination had the highest AUC (0.93), potentially improving PACG detection [109].

Limited literature is currently available about the potential association of Corvis ST measurements and glaucoma severity with varying results. A study by Wu and colleagues investigated the relationship between corneal biomechanics as measured by Corvis ST and visual field changes. They reported that the shorter the WEM, the worse the MD (*p* = 0.02) and the higher the pattern’s standard deviation (PSD) (*p* = 0.03) in NTG. However, these associations were not found in HTG [33]. Similarly, Bolivar and colleagues reported no significant association between Corvis ST parameters and visual field MD or PSD in POAG patients [110]. It should be noted that both studies were performed on treatment-naïve patients [33,110]. In contrast, Hirasawa and colleagues reported significant correlations between A1V and A2T and glaucomatous visual field defects [111]. Vinciguerra and colleagues reported that in POAG eyes (HTG and NTG), there was a significant negative correlation between DA ratio (*p* = 0.01) and inverse concave radius (*p* = 0.02) and MD and a significant positive correlation between SP-A1 (*p* = 0.01) and SP-HC (*p* = 0.03) and MD [97]. There was also a significant association between PSD and Corvis ST measurements. Notably, both Hirasawa and Vinciguerra included patients on glaucoma medications, including PGA, which may have confounded the results [97,111]. The contradictory results may be explained at least in part by the inclusion of treatment-naïve patients in the first two studies and treated patients in the last two. A prospective study by Qassim and colleagues investigated the correlation between SP-A1 and the risk of glaucoma progression in 228 glaucoma suspects. They demonstrated that the higher the SP-A1, the faster the rate of RNFL and RGC loss (*p* < 0.001). They reported that patients with higher SP-A1 and lower CCT had 2.9-folds increased risk of RNFL loss > 1 µm/year (*p* = 0.006) [112].

## 6. Prostaglandin Analogues-Induced Corneal Biomechanical Changes

There is growing evidence about the effect of PGA on ocular biomechanics related to the mechanism of increased uveoscleral outflow. Several studies have demonstrated that topical PGA therapy may result in increased matrix metalloproteinase expression and decreased expression of tissue inhibitors of metalloproteinases, which subsequently results in structural changes in the outer coat of the eye. These changes are hypothesized to reduce the ocular stiffness and increase permeability to aqueous outflow, reducing the IOP [31,113,114,115]. This same effect could result in corneal biomechanical changes that may confound the accuracy of serial IOP readings.

There is a controversy about the effect of topical PGA on CH and CRF. A study by Tsikripis and colleagues evaluated 108 POAG eyes on latanoprost with or without timolol. They reported that the mean CH significantly increased in both groups, whereas CRF did not change [116]. Another study by Meda and colleagues evaluated 70 eyes of 35 patients treated with long-term PGA. The PGA therapy was stopped for six weeks in one eye of each patient. They reported that cessation of PGA increased CH and CRF [117].

Using Corvis ST, Wu and colleagues compared the changes in corneal biomechanics of treatment-naïve POAG patients versus POAG under chronic PGA therapy (for at least two years) versus normal controls. Although they concluded that long-term PGA therapy induces deformational changes based on the significant increase in the DA, this may be related to the fact that DA inversely correlates with IOP. Hence, the measured increase in DA may have been because of decreased IOP with PGA therapy [98,108,118]. Similarly, Sanchez-Barahona and colleagues reported that three months of PGA therapy in treatment-naïve POAG patients resulted in significant changes in corneal biomechanics based on significant changes in A1T (*p* = 0.001), A2T (*p* = 0.001), and DA (*p* = 0.0003) [119]. However, all three parameters are strongly affected by IOP, and changes may also be explained by IOP reduction with PGA therapy. Therefore, future studies need to evaluate the PGA-induced biomechanical changes using more recent parameters that are more heavily affected by stiffness than IOP changes, such as SP-A1, SP-HC, SSI, and DA ratio [32,120].

On the other hand, Zheng and colleagues [121] studied the effect of travoprost on rabbit cornea biomechanics and showed that topical PGA resulted in softer corneas with decreased resistance to deformation. This raises the question of the actual IOP-lowering effect of the PGA therapy since, theoretically, part of it may be related to measurement artifacts with softer corneas that may underestimate IOP measurement [31]. However, further studies are needed to better assess the IOP-lowering effect of PGA independent of its effect on corneal biomechanics. Another ex vivo study investigated the effects of travoprost and tafluprost on rabbit corneas. The authors demonstrated a significant decrease in the tangent modulus by almost 30% and increased interfibril spacing after PGA therapy [122].

## 7. Effect of Other Topical Anti-Glaucoma Medications on Corneal Biomechanics

Limited literature is available about the effect of IOP-lowering medications other than PGA on corneal biomechanics. One study by Aydemir and colleagues [123] reported that there was a statistically significant difference in the CH between patients on benzalkonium chloride containing brimonidine (8.77 mmHg) compared to healthy controls (10.26 mmHg) (*p* = 0.02). However, there was no difference in the CH or CRF between purite-containing brimonidine and the control group. This study highlights the effect of preservative agents on corneal biomechanics. 

## 8. Conclusions

The biomechanical characterization of the cornea has emerged as an exciting frontier in glaucoma diagnosis and management. However, the limitations of existing methods and the weak to moderate agreement between the reported parameters limited their widespread use in clinical practice. Additionally, studies of corneal biomechanics in glaucoma are further limited by their inclusion of glaucoma subjects taking topical PGA, which may alter corneal biomechanics, leading to contradicting results. Furthermore, some studies lack proper patient stratification and sometimes misinterpret results due to reported factors that are confounded by IOP changes. It is important to note that corneal biomechanical properties are dynamic metrics and can change over time with age, corneal trauma, or surgery. 

There is a clear need for a more robust measure of corneal biomechanics that can more accurately determine the modulus of elasticity. New devices such as BM represent a promising, novel approach to evaluating corneal biomechanical properties in glaucoma patients independent of IOP. 

## Figures and Tables

**Figure 1 bioengineering-10-01108-f001:**
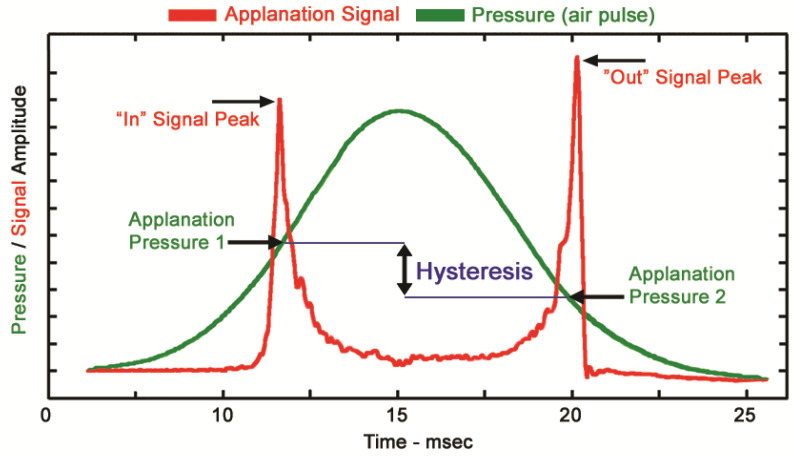
The applanation signal and air pulse pressure diagram was obtained over the course of 1 measurement. Applanation pressure 1 was the pressure at which the cornea reached a specific applanation state on inward movement, and applanation pressure 2 was the pressure at which the cornea passed through this applanation state on outward movement. The difference between these two pressures was the “corneal hysteresis” parameter, which was the main output by the machine. The image was obtained from http://www.reichert.com/ (accessed on 23 March 2023).

**Figure 2 bioengineering-10-01108-f002:**
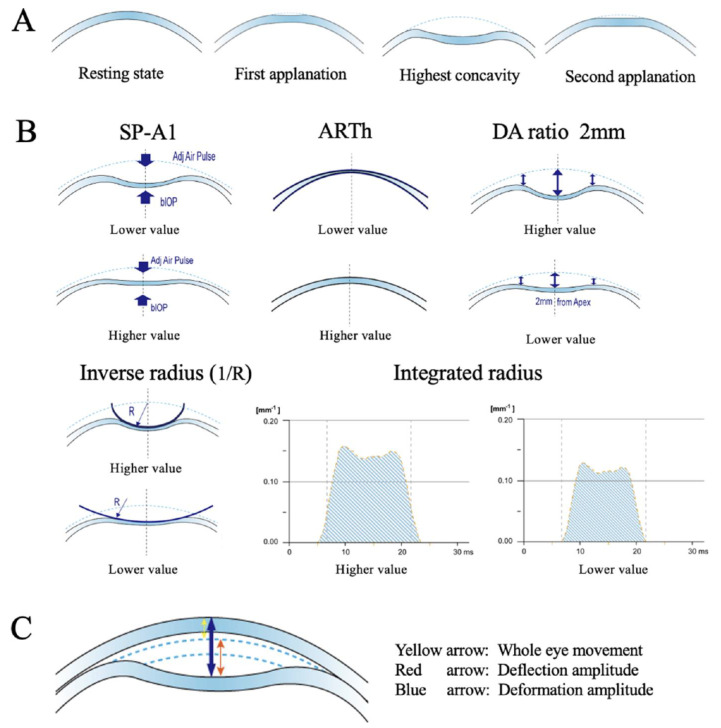
Schematic diagrams of the ocular biomechanical parameters provided by Corvis ST. (**A**) Cornea deformation during the Corvis ST measurement. From left to right: resting state before the measurement; first applanation; highest concavity; and second applanation. (**B**) Graphs illustrating SP-A1, ARTh, DA ratio 2 mm, inverse radius, and integrated radius. Lower values of SP-A1 and ARTh indicate a more deformable cornea, whereas higher values of DA ratio 2 mm, inverse radius, and integrated radius indicate a more deformable cornea. (**C**) Correlation between deformation amplitude and whole eye movement. Deformation amplitude is the sum of whole eye movement and pure corneal deformation named deflection amplitude. Yellow arrow: whole eye movement; Red arrow: deflection amplitude; and Blue arrow: deformation amplitude. Reprinted with permission from Wu N, Chen Y, and Sun X. Association Between Ocular Biomechanics Measured With Corvis ST and Glaucoma Severity in Patients With Untreated Primary Open Angle Glaucoma. Transl Vis Sci Technol. 2022;11(6):10. doi:10.1167/tvst.11.6.10 [33].

**Table 1 bioengineering-10-01108-t001:** Studies evaluating corneal hysteresis in glaucoma/ocular hypertension (OHT) patients compared to healthy controls using the ocular response analyzer.

Study	Number of Patients		Prostaglandin Therapy in the Glaucoma Group	Parameters That Were Significantly Different between Study Groups
	Glaucoma/OHT	Healthy controls		
Kirwan and colleagues, 2006 [49]	8 (CG)	42	Not reported	- CG eyes had a statistically significantly lower mean CH (6.3 mmHg) compared to normal controls (12.5 mmHg)
Sullivan-Mee and colleagues, 2008 [74]	99 (primary glaucoma) 58 (OHT) 70 (GS)	71	- Glaucoma: 33% use PGA alone and 40% used PGA with adjunct - OH group: 31% used PGA alone and 7% used PGA with adjunct	- Glaucoma group had a significantly lower mean CH (8.1 mmHg) compared to OHT (8.9 mmHg), GS (8.9 mmHg), and normal (9.7 mmHg). - OHT group had a significantly higher CRF (10.2 mmHg) compared to glaucoma (8.3 mmHg), GS (8.5 mmHg), and normal (9.2 mmHg)
Mangouritsas and colleagues, 2009 [51]	108 (POAG)	74	- 42.6% were treated with one medication and 57.4% were treated with >1 medication. However, medication details were not reported	- POAG eyes had a statistically significantly lower mean CH (8.9 mmHg) compared to normal controls (10.9 mmHg)
Sun and colleagues, 2009 [52]	40 (unilateral CPACG)	40	- 8/40 used PGA	- CPACG eyes had a statistically significantly lower mean CH (6.8 mmHg) compared to the fellow eyes (10.5 mmHg) and normal controls (10.5 mmHg)
Abibtol and colleagues, 2010 [53]	58 (OAG-HTG)	75	- All patients were treated with glaucoma medication. However, no details were reported	- Glaucomatous eyes had a statistically significantly lower mean CH (8.7 mmHg) compared to normal controls (10.4 mmHg)
Ayala, 2011 [63]	30 (POAG) 30 (PXG)	30	- POAG and PXG patients were on glaucoma medications. However, no details were reported	- CH was significantly lower in PXG compared to normal subjects and POAG - No significant difference in CH between normal controls and POAG
Narayanaswamy and colleagues, 2011 [54]	162 (POAG-HTG and NTG) 131 (PACG)	150	- Patients using medications were not excluded. However, details were not reported	- After adjusting for age, sex, and IOP, CH was significantly lower in PACG (9.4 mmHg) compared to normal controls (10.1 mmHg) - No difference in CH between POAG and normal controls
Kaushik and colleagues, 2012 [55]	36 (POAG-HTG) 18 (POAG-NTG) 101 (GS) 38 (OHT) 59 (PACD)	71	- Patients on any topical ophthalmic treatment were excluded from the study	- CH was significantly lower in HTG (7.9 mmHg) and NTG (8.0 mmHg) compared to normal controls (9.5 mmHg) - CRF was lowest In NTG (7.8 mmHg) and highest in HTG (11.1 mmHg)
Grise-Dulac and colleagues, 2012 [56]	38 (POAG-HTG) 14 (NTG) 27 (OHT)	22	- Not reported	- NTG eyes had a statistically significantly lower mean CH (9.8 mmHg) compared to normal controls (11.05 mmHg) - NTG eyes had a statistically significantly lower mean CRF (9.5 mmHg) compared to normal controls (11.00 mmHg) and HTG (11.1 mmHg)
Derty-Morel and colleagues, 2012 [57]	59 (POAG)	55	- Not reported	- African healthy controls and POAG patients had a significantly lower CH compared to Caucasian normal and POAG patients
Morita and colleagues, 2012 [58]	83 (NTG)	83	- Not reported	- NTG eyes had a statistically significantly lower mean CH and CRF (9.2 mmHg and 8.9 mmHg, respectively) compared to normal controls (10.8 mmHg and 10.6 mmHg, respectively)
Cankaya and colleagues, 2012 [59]	64 (PEX) 78 (PXG)	102	- 12/78: PGA alone - 34/78: PGA and other medications	- PXG eyes had a statistically significantly lower mean CH (6.9 mmHg) compared to normal controls (9.4 mmHg) and PEX eyes (8.5 mmHg) - CRF was not significantly different in PXG eyes (9.5 mmHg) compared to the control group (9.8 mmHg) and PEX (9.3 mmHg)
Beyazyildiz and colleagues, 2014 [60]	66 (POAG) 46 (PXG)	50	- POAG: 54% - PXG: 63%	- PXG eyes had a statistically significantly lower mean CH (7.6 mmHg) compared to normal controls (9.6 mmHg) and POAG eyes (9.1 mmHg) - CRF was significantly lower in PXG eyes (9.0 mmHg) compared to the control group (9.8 mmHg) and POAG (10.1 mmHg)
Shin and colleagues, 2015 [61]	97 (POAG-NTG)	89	- 47/97	- NTG eyes had a statistically significantly lower mean CH and CRF (9.9 mmHg and 9.7 mmHg, respectively) compared to normal controls (10.5 mmHg and 10.5 mmHg, respectively)
Hussnain and colleagues, 2015 [62]	322 (POAG)	1418	- Not reported	- POAG eyes had a statistically significantly lower mean CH (9.5 mmHg) compared to normal controls (9.9 mmHg)
Yazgan and colleagues, 2015 [64]	43 eyes (PEX) 30 eyes (PXG)	45 eyes	- 17/30	- PXG eyes had a statistically significantly lower mean CH (6.8 mmHg) compared to normal controls (10.3 mmHg) and PEX eyes (8.2 mmHg) - CRF was significantly higher in the control group (10.3 mmHg) compared to PEX (7.9 mmHg) and PXG (7.9 mmHg)
Dana and colleagues, 2015 [65]	37 eyes (POAG)	21 eyes	- Not reported	- POAG eyes had a lower CH (9.8 mmHg) and CRF (10.3 mmHg) compared to control eyes (11.0 and 11.6, respectively)
Pillunat and colleagues, 2016 [66]	48 (POAG-HTG) 38 (POAG-NTG) 18 (OHT)	44	- Patients were on topical medications; however, details were not reported	- POAG eyes had a statistically significantly lower mean CH (8.9 mmHg) and CRF (9.07 mmHg) compared to OHT (CH: 10.2 mmHg, CRF: 10.7 mmHg) and normal controls (CH: 9.7 mmHg, CRF: 10.2 mmHg)
Perucho-Gonzalez and colleagues, 2016 [67]	78 (PCG)	53	- Not reported	- PCG eyes had a statistically significant lower CH (8.5 vs. 11.3 mmHg) and CRF (9.8 vs. 11.02 mmHg) compared to controls
Perucho-Gonzalez and colleagues, 2017 [68]	66 (PCG)	94	- Not reported	- PCG eyes had a statistically significant higher AME (9.0 vs. 3.2), PAE (3.1 vs. 0.9), and PME (30.8 vs. 7.5) compared to controls - PCG eyes had a statistically significant lower CH (8.5 vs. 11.1 mmHg) and CRF (9.9 vs. 10.7 mmHg) compared to controls
Park and colleagues, 2018 [69]	95 (POAG-NTG)	93	- Patients on glaucoma medications were excluded	- NTG eyes had significantly lower CH (10.5 mmHg) and CRF (10.1 mmHg) compared to normal controls (10.8 and 10.6 mmHg, respectively)
Potop and colleagues, 2020 [73]	79 eyes (POAG regardless of IOP) 68 eyes (OHT)	67 eyes	- Patients on glaucoma medications were not excluded	- POAG eyes had lower CH (8.5 mmHg) compared to OHT (9.6 mmHg) and normal controls (11.7 mmHg)
Aoki and colleagues, 2021 [70]	68 (POAG)	68	- Patients on glaucoma medications were not excluded	- CH was significantly lower in POAG (8.9 mmHg) compared to normal eyes (9.9 mmHg)
Rojananuangnit, 2021 [71]	272 (POAG) 143 (NTG) 48 (PACG) 30 (OHT)	465	- POAG: 414/434 eyes - NTG: 141/143 eyes - PACG: 46/74 eyes - OHT: 24/44 eyes	- CH in OHT (10.1 mmHg) was significantly higher than POAG (8.74) and PACG (9.09 mmHg) - No statistically significant difference in CH between OHT (10.1 mmHg) and NTG (9.5 mmHg) - The CH was significantly lower in the glaucoma groups compared to normal controls
Del Buey-Sayas and colleagues, 2021 [72]	491 (Glaucoma or GS)	574	- Not reported	- CH in glaucoma patients (9.6 mmHg) is lower than in the control group (10.7 mmHg) and all forms of GS

CG: congenital glaucoma, GS: glaucoma suspect, OAG: open-angle glaucoma, POAG: primary open-angle glaucoma, CPACG: chronic primary angle-closure glaucoma, HTG: high-tension glaucoma, NTG: normal-tension glaucoma, PCG: primary congenital glaucoma, COAG: chronic open-angle glaucoma, PEX: pseudoexfoliation syndrome, PXG: pseudoexfoliation glaucoma, PACD: primary angle-closure disease, PACG: primary angle-closure glaucoma, CH: corneal hysteresis, CRF: corneal resistance factor, AME: anterior maximum elevation, PAE: posterior apex elevation, and PME: posterior maximum elevation.

**Table 2 bioengineering-10-01108-t002:** Studies evaluating corneal biomechanics in glaucoma/ocular hypertension (OHT) patients compared to healthy control using Corvis ST.

Study	Number of Patients		Prostaglandin Therapy in the Glaucoma Group	Parameters Evaluated *	Parameters That Were Significantly Different between Study Groups	Conclusion
	Glaucoma/OHT	Healthy controls				
Leung and colleagues, 2013 [93]	101 glaucomatous eyes 39 glaucoma suspect eyes	40	PGAs were used, but the exact number of patients on PGA was not reported	5 parameters - A1L - A2L - A1V - A2V - DA	None of the five factors were statistically significantly different between both groups	
Salvetat and colleagues, 2015 [88]	85 (POAG)	79	33/87 patients	10 parameters - A1T - A1L - A1V - A2T - A2L - A2V - HCT - DA - PD - Radius HC	- A1T was higher in the POAG group (*p* = 0.007). - The A1V (*p* = 0.04), A2T (*p* < 0.001), A2V (*p* = 0.014), and DA HC (*p* < 0.001) were lower in the POAG group.	POAG eyes have less deformable corneas than controls
Wang and colleagues, 2015 [89]	37 (POAG-HTG)	36	Patients on glaucoma medications were not excluded from the study	10 parameters - A1T - A1L - A1V - A2T - A2L - A2V - HCT - DA - CCR - PD	- A1T, A2V, and PD were higher in the POAG (*p* < 0.05). - DA, A1V, and A2T were lower in the POAG (*p* < 0.05).	POAG eyes have less deformable cornea compared to controls
Coste and colleagues, 2015 [90]	37 (COAG)	19	Not reported	7 parameters - DA - A1T - A2T - HCT - A1L - A2L - Corneal velocity	- DA was significantly lower in the COAG group - HCT was significantly shorter in the COAG group	Corneal deformation is lower in glaucomatous patients compared to controls
Lee and colleagues, 2016 [94]	34 (POAG-HTG) 26 (POAG-NTG)	61	79.5% were on glaucoma medications	10 parameters - A1T - A1L - A1V - A2T - A2L - A2V - HCT - DA - PD - Radius HC	- A2V (*p* = 0.001) and PD (*p* = 0.005) were greater in the glaucoma group - HCT was shorter in the glaucoma group (*p* = 0.002)	
Tian and colleagues, 2016 [95]	42 (POAG-HTG)	60	34/42 patients	10 parameters - A1T - A1L - A1V - A2T - A2L - A2V - HCT - DA - PD - Radius HC	- DA was significantly lower in the POAG (*p* < 0.001). - A1V, A2T, and PD were lower in the POAG group	Corneal deformation is lower in glaucomatous patients compared to controls
Wu and colleagues, 2016 [108]	69	19	35/69 (treatment naïve) 34/69 (at least 2 years of PGA therapy)	10 parameters - A1T - A1L - A1V - A2T - A2L - A2V - DA - PD - Radius HC - HCT	- After adjusting for age, gender, IOP, CCT, axial length, and corneal curvature, DA was significantly lower in treatment-naïve POAG compared to POAG on PGA therapy and controls	- Treatment-naïve POAG eyes have less deformable corneas compared to patients on PGA therapy
Jung and colleagues, 2017 [96]	136 (OAG)	75	82/136 patients	9 parameters - A1L - A1V - A2L - A2V - DA - PD - Radius HC - WEM - DFA	- DA was smaller compared to controls (*p* = 0.03)	Corneal deformation is lower in glaucomatous eyes compared to controls
Hong and colleagues, 2019 [91]	80 (POAG-NTG)	155	76% were on glaucoma medications but they did not specify the number	10 parameters - A1T - A1L - A1V - A2T - A2L - A2V - HCT - DA - PD - Radius HC	- A1V was significantly higher in the NTG group	NTG has more deformable corneas compared to controls
Miki and colleagues, 2019 [92]	75 (POAG-medically controlled)	47	Mean number of topical medications was 1.8 ± 1.2. However, % of eyes that used prostaglandin analogues was not specified	8 parameters - A1T - A1V - A2T - A2V - HC DFA - PD - Radius HC - WEM	- Glaucoma was negatively correlated with A1T, A2T, radius HC, and WEM	- Eyes with medically controlled POAG are more deformable compared to normal controls
Pillunat and colleagues, 2019 [39]	70 (POAG-NTG)	70	115/140 eyes	They used five parameters (DA ratio progression, HCT, Pachymetry slope, biomechanically corrected IOP, Pachymetry) to calculate Dresden BGF	- The BGF was statistically higher in the NTG (0.67) compared to normal controls (0.33) (*p* < 0.001). - DA ratio progression was higher and HCT was shorter in NTG compared to controls	- Using a cut-off of 0.5 BGF, NTG can be differentiated from normal controls and correctly classified in 76% of eyes - NTG eyes may have stiffer corneas with reduced viscoelastic capability
Vinciguerra and colleagues, 2020 [97]	41 (POAG-HTG) 33 (POAG-NTG) 45 (OHT)	37	37/41 patients (POAG-HTG) 23/33 patients (POAG-NTG) 31/45 patients (OHT)	4 parameters - SP-A1 - SP-HC - Inverse concave radius - DA ratio	- SP-A1 and SP-HC were significantly lower in NTG compared to the other three groups - Inverse concave radius and DA ratio were significantly higher in NTG compared to the other 3 groups	NTG eyes have a more deformable cornea compared to HTG, OHT, and controls
Miki and colleagues, 2020 [98]	35 (POAG-NTG)	35	0 (All patients were treatment-naïve)	10 parameters - A1T - A1V - A2T - A2V - HC DFA - PD - Radius HC - DA ratio 1 mm - Integrated radius - WEM Max	- A1T, A2T, and radius HC were significantly smaller in NTG compared to controls - PD, DA 1 mm, and integrated radius were significantly larger in NTG compared to controls	Corneas of untreated NTG eyes are more deformable compared to controls
Pradhan and colleagues, 2020 [100]	29 (POAG including NTG) 32 (PXG)	33	0 (All patients were treatment-naïve)	7 parameters - A1L - A1V - A2L - A2V - DA - PD - Radius HC	- After adjusting for IOP, there was no difference in any parameter between the three groups	No difference in corneal deformability between POAG, PXG, and controls
Pradhan and colleagues, 2020 [101]	27 (PXG) 14 (PXF + OHT) 29 (PXF)	32	0 (All patients were treatment-naïve)	7 parameters - A1L - A1V - A2L - A2V - DA - PD - Radius HC	- DA and corneal velocities were significantly lower in PXG and PXF + OHT compared to PXF and normal controls - After adjusting for IOP and age, there was no difference in any parameter between the four groups	No difference in corneal deformability between PXG, PXF, PXF + OHT, and controls
Jung and colleagues, 2020 [99]	46 (POAG-HTG) 54 (POAG-NTG)	61	32/46 in HTG 38/54 in NTG	7 parameters - A1L - A1V - A2L - A2V - PD - DA - Radius HC	- A1V and DA were smaller in HTG compared to NTG and controls - Radius HC was larger in HTG compared to controls	Eyes with POAG-HTG have less deformable corneas compared to NTG and controls
Aoki and colleagues, 2021 [70]	68 (POAG)	68	56/68	BGF	- No statistical difference in BGF between POAG eyes (0.61) and normal controls (0.51)	- BGF is not useful in differentiating POAG eyes from normal controls
Wei and colleagues, 2021 [102]	45 (POAG-HTG) 49 (POAG-NTG)	50	Several glaucoma patients were on PGA, but they did not report a specific number	19 parameters - Max inverse concave radius - DAR 2mm - DAR 1mm - Integrated radius - SP-A1 - A1-DFL - HC-DFL - A2-DFL - A1-DFA - HC-DFA - A2-DFA - DFA Max - WEM - A1-DF Area - HC-DF Area - A2-DF Area - A1-dDFL - A2-dDFL - dDFL Max - HC-dDFL	- Maximum inverse concave radius and DAR (1 and 2 mm) were significantly higher in NTG eyes compared to controls - Integrated radius and DAR 2 mm were significantly higher in NTG compared to HTG - SPA-1 was significantly lower in NTG compared to HTG - No significant difference in any of the parameters between HTG and normal controls	- NTG eyes have more deformable corneas compared to HTG and controls - No difference in corneal deformability between HTG and controls
Silva and colleagues, 2022 [103]	61 eyes (POAG) 32 eyes (Amyloidotic glaucoma) 37 eyes (OHT)	53 eyes	72% (POAG) 59% (Amyloidotic glaucoma) 59% (OHT)	14 parameters - A1T - A1V - A2T - A2V - A1-DFL - A2-DFL - PD - Radius HC - DA HC - HC-DFA - SSI - SP-A1 - DA ratio - Integrated radius	- Eyes with OHT had significantly higher SPA-1 compared to POAG, and SSI compared to amyloidotic glaucoma - Eyes with amyloidotic glaucoma had lower HC-DFA and higher integrated radius compared to controls	- Eyes with OHT have less deformable corneas compared to POAG, Amyloidotic glaucoma, and controls
Zarei and colleagues, 2022 [104]	66 eyes (POAG-HTG) 21 eyes (POAG-NTG) 26 eyes (PXG) 46 eyes (PACG)	70 eyes		31 parameters - A1T - A1V - A2T - A2V - HCT - PD - Radius HC - A1-DA - HC-DA - A2-DA - A1-DFL - HC-DFL - A2-DFL - A1-DFA - HC-DFA - A2-DFA - DFA Max - WEM Max - A1-DF area - HC- DF area - A1-dArc length - HC-dArc length - A2-dArc length - dArc length Max - DA ratio Max - ARTh - Integrated radius - SP-A1 - SSI - CBI	- Radius indices were lower in HTG, NTG, and PXG compared to controls - Max inverse radius and integrated radius were higher in PACG compared to controls	Altered corneal biomechanics in different types of glaucoma
Xu and colleagues, 2022 [107]	113 (POAG-HTG) 108 (POAG-NTG)	113	47/113 (POAG-HTG) 42/108 (POAG-NTG)	5 parameters ** - DA - DA ratio - Integrated radius - SPA-1 - SSI	- DA was higher in the NTG compared to HTG (*p* = 0.03) but not when compared to controls (*p* = 0.93) - No significant difference between three groups in DA ratio, integrated radius, SP-A1, and SSI measurements	Based on Corvis ST *** results, NTG eyes have more deformable corneas compared to HTG but not when compared to controls
Wu and colleagues, 2022 [33]	55 (POAG-HTG) 47 (POAG-NTG)	51	0 (All patients were treatment-naïve)	13 parameters - A1T - A1V - A2T - A2V - HCT - DA - PD - Radius HC - SP-A1 - Integrated radius - ARTh - DA ratio 2 mm - WEM	- DA was significantly higher and A1T and HC time were significantly lower in NTG, and HTG compared to normal controls - Comparing NTG and HTG, only A1V was significantly different being lower in HTG	NTG eyes have more deformable corneas compared to HTG and normal controls - HTG eyes have more deformable corneas compared to normal controls
Vieira and colleagues, 2022 [106]	70 (POAG-HTG) 16 (PXG) 23 (OHT)	37	All glaucoma patients and 92.9% of OHT were medically treated. However, details were not reported	8 parameters - A1L - A1V - A2L - A2V - PD - Radius HC - DA - CSI	- OHT has significantly higher A1L, A2V and lower A1V compared to POAG and PXG	Eyes with OHT have stiffer corneas compared to healthy controls, POAG, PXG
Halkiadakis and colleagues, 2022 [105]	30 (POAG-HTG) 25 (OHT)	25	POAG and OHT were medically treated but details were not reported	15 parameters - A1T - A2T - A1L - A2L - A1V - A2V - HCT - Radius HC - HCC - DA - DA 2mm - SP-A1 - Inverse concave radius	- A2T was the only parameter that was statistically significantly different between groups being shorter in POAG (*p* = 0.048)	- Corneas of POAG may have altered viscoelasticity based on reduced A2T

POAG: primary open-angle glaucoma, HTG: high-tension glaucoma, NTG: normal-tension glaucoma, COAG: chronic open-angle glaucoma, OAG: open-angle glaucoma, PXF: pseudoexfoliation syndrome, PXG: pseudoexfoliation glaucoma, PACG: primary angle-closure glaucoma, PGA: prostaglandin analogues, A1T: first applanation time, A1L: length of the flattened cornea in A1, A1V: first applanation velocity, A2T: second applanation time, A2L: length of the flattened cornea in A2, A2V: second applanation velocity, DA: deformation amplitude, HCT: time from start until cornea reaching the highest concavity, PD: peak distance, HC: highest concavity, CCR: central curvature radius of the cornea at the highest concavity, WEM: whole eye movement, DFA: deflection amplitude, DF area: area displaced as a result of corneal deformation in the horizontal section analyzed, IOP: intraocular pressure, SP-A1: stiffness parameter at first applanation, SP-HC: stiffness parameter at highest concavity, HCC: central curvature radius at the highest concavity, Max: maximum, mm: millimeter, dDFL: delta deflection arc length, ARTh: Ambrosio relational thickness to the horizontal profile, CBI: Corvis biomechanical index, SSI: stress–strain index, CSI: concavity shape index, and BGF: biomechanical glaucoma factor. *: Apart from intraocular pressure or central corneal thickness. **: There was no separate statistical analysis for HTG compared to NTG. ***: In addition to Corvis ST, a corneal indentation device was also used to evaluate corneal stiffness and showed that the corneal stiffness in NTG was lower than that of HTG (*p* = 0.001) and controls (*p* = 0.023).

## Data Availability

Data used in this review article are publicly available.

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
