# Peer review of "Corneal Biomechanical Measures for Glaucoma: A Clinical Approach"

_bioengineering, 2023, doi:10.3390/bioengineering10101108_

Round 1
Reviewer 1 Report
Major comments:
The introduction is poorly written. It needs to be refined and to discuss different parameters measured from Ocular Response Analyzer (ORA; Reichert; NY; USA), and the Corneal Visualization Scheimpflug Technology, and their interpretation and Brillouin microscopy (BM). How the clinicians could interpret these parameters differently; so it can highlight the novelty of your research.
Limitations of each techniques should be discussed in in the relevant sections with examples from the literature.
strain = deformation/original shape can you give more measurable terms
Some definition needs to be revised for example Stiffness is the ability of a material to return to its original shape after a force is applied to it. This is the definition of elasticity; However, Stiffness is the measure of resistance to deformation with stress.
Again, viscosity means that part of the applied force is lost to internal friction as heat. You need to revise this definition and find a supporting reference. Viscosity (force. Time/surface area) is the resistance of a fluid (liquid or gas) to a change in shape relative to one another. Viscosity is a measure of a fluid's resistance to flow. The SI unit of viscosity is poiseiulle (PI). Its other units are newton-second per square metre (N s m-2) or pascal-second (Pa s.).
PGA therapy since theoretically part of it may be related to measurement 446 artifact. This is a strong statement and needs to be substantiated with references from the literature.
Long term trials of PGA and clinical outcomes on visual fields and optic nerve damage needs to be discussed to support your hypothesis
How the literature describes other antiglaucomic treatments and their effect on biomechanics of the cornea.
The authors are strongly advised to have a separate section for conclusion, limitations and directions for future research.
None
Reviewer 2 Report
This paper outlines the importance of evaluating corneal biomechanics in various ocular diseases, particularly glaucoma, while highlighting the challenges and current limitations in the field. The paper provides valuable information, but a few suggestions can be considered to enhance its clarity and structure:
Introduction and Context: Start by introducing the significance of assessing corneal biomechanics, especially in the context of glaucoma. Explain why understanding corneal biomechanics can offer insights into disease pathogenesis, diagnosis, and management.
Clear Objectives: Explicitly state the objectives of the review. Outline what readers can expect to gain from the review, such as an overview of recent evidence, challenges faced, and recommendations for future studies.
Clarity on Devices: Provide a brief explanation of the two devices used for evaluating corneal biomechanics (Ocular Response Analyzer and Corneal Visualization Scheimpflug Technology). Explain their roles in assessing corneal biomechanics and why there is variability in the parameters they report.
Challenges with Study Inclusion: Elaborate on why including glaucoma subjects using topical prostaglandin analogues could alter corneal biomechanics. Discuss how such alterations might impact the interpretation of study results.
Research Gaps and Future Directions: Highlight the specific limitations of current studies investigating corneal biomechanics in glaucoma patients. Provide insights into how these limitations could be addressed in future research. This section should emphasize the actionable steps that researchers can take to improve study design and interpretation.
By incorporating these suggestions, the paper will provide a clearer, more organized, and impactful overview of the review's content, objectives, and contributions to the field of corneal biomechanics in glaucoma research.
The writing is clear, and the sentences are generally well-structured.
Round 2
Reviewer 1 Report
The authors have satisfactorily addressed my comments. One minor comment is that:
The authors have suggested that the 215 optimal model for calculating CH using Corvis ST parameters is: CH= -76.3+4.6xA1T + 216 1.9xA2T + 3.1xDA + 0.016xCCT. Please rewrite the sentence so it reads passive words and remove the word authors.
Minor English edits are required.
Author Response
See attached document
